# Pulmonary function following hyperbaric oxygen therapy: A longitudinal observational study

Connor T. A. Brenna[1], Shawn Khan[2], George Djaiani[3], Darren Au[4],
Simone Schiavo[1,3,4], Mustafa Wahaj[3], Ray Janisse[3], Rita Katznelson[1,3,4]*

**1** Department of Anesthesiology & Pain Medicine, University of Toronto, Toronto, Ontario, Canada, **2** Faculty of Medicine, University of Toronto, Toronto, Ontario, Canada, **3** Hyperbaric Medicine Unit, Toronto General Hospital, Toronto, Ontario, Canada, **4** Department of Anesthesia and Pain Management, University Health Network, Toronto, Ontario, Canada

* rita.katznelson@uhn.ca

**Data Availability Statement:** The source data from our study cannot be shared publicly, in full, because of an ethical restriction levied by our institutional research ethics board. This is because the data contains sensitive information from

## Abstract

Hyperbaric oxygen therapy (HBOT) is known to be associated with pulmonary oxygen toxicity. However, the effect of modern HBOT protocols on pulmonary function is not completely understood. The present study evaluates pulmonary function test changes in patients undergoing serial HBOT. We prospectively collected data on patients undergoing HBOT from 2016–2021 at a tertiary referral center (protocol registration NCT05088772). Patients underwent pulmonary function testing with a bedside spirometer/pneumotachometer prior to HBOT and after every 20 treatments. HBOT was performed using 100% oxygen at a pressure of 2.0–2.4 atmospheres absolute (203–243 kPa) for 90 minutes, five times per week. Patients' charts were retrospectively reviewed for demographics, comorbidities, medications, HBOT specifications, treatment complications, and spirometry performance. Primary outcomes were defined as change in percent predicted forced expiratory volume in one second ($FEV_1$), forced vital capacity (FVC), and forced mid-expiratory flow ($FEF_{25-75}$), after 20, 40, and 60 HBOT sessions. Data was analyzed with descriptive statistics and mixed-model linear regression. A total of 86 patients were enrolled with baseline testing, and the analysis included data for 81 patients after 20 treatments, 52 after 40 treatments, and 12 after 60 treatments. There were no significant differences in pulmonary function tests after 20, 40, or 60 HBOT sessions. Similarly, a subgroup analysis stratifying the cohort based on pre-existing respiratory disease, smoking history, and the applied treatment pressure did not identify any significant changes in pulmonary function tests during HBOT. There were no significant longitudinal changes in $FEV_1$, FVC, or $FEF_{25-75}$ after serial HBOT sessions in patients regardless of pre-existing respiratory disease. Our results suggest that the theoretical risk of pulmonary oxygen toxicity following HBOT is unsubstantiated with modern treatment protocols, and that pulmonary function is preserved even in patients with pre-existing asthma, chronic obstructive lung disease, and interstitial lung disease.

patient's medical charts (e.g., birth dates and personal health information). Taken together, this information may allow for the identification of individual study participants. We offer that an anonymized minimal data set can be prepared in aggregate and made available upon reasonable request via email to the study's first author (connor.brenna@mail.utoronto.ca) or the Hyperbaric Medicine Unit, Toronto General Hospital, Toronto, Ontario, Canada (hyperbaricmedicineunit@uhn.ca).

**Funding:** CTAB gratefully acknowledges the William S. Fenwick Research Fellowship received from the University of Toronto Temerty Faculty of Medicine in support of this study.

**Competing interests:** I have read the journal's policy and the authors of this manuscript have the following competing interests: RK is a shareholder in the Rouge Valley Hyperbaric Medical Center, Toronto, ON. This does not alter our adherence to PLOS ONE policies on sharing data and materials.

**Abbreviations:** HBOT, hyperbaric oxygen therapy; $O_2$, oxygen; POT, pulmonary oxygen toxicity; ATA, atmosphere absolute; PFT, pulmonary function test; $FEV_1$, percentage of predicted forced expiration volume in one second; FVC, percentage of predicted forced vital capacity; $FEF_{25-75}\%$, percentage of predicted mid-expiratory flow; DC, diffusion capacity; PEF, peak expiratory flow; RV, residual volume; TLC, total lung capacity; VC, vital capacity; UPTD, unit of pulmonary toxic dose.

## Introduction

Hyperbaric oxygen therapy (HBOT) has been recognized as a valuable intervention for a variety of acute and chronic conditions (S1 Table) [1, 2]. Treatment protocols include repeated sessions of exposure to 100% oxygen ($O_2$) at 1.3–2.8 atmospheres absolute (ATA) or 132–284 kPa for a predetermined amount of time per session, with a variable number of sessions per week and up to 60 total sessions depending on the indication. Although individual treatments may incorporate air breaks to avoid potential pulmonary and neurological $O_2$ toxicity, the cumulative effect of multiple longitudinal sessions of HBOT on pulmonary function is not completely understood.

Pulmonary oxygen toxicity (POT) related to the high partial pressure of $O_2$ in the alveoli may impair respiratory function [3]. Although the mechanisms of POT (also called the Lorrain Smith effect) [4] remain unclear, the increased production of reactive $O_2$ species during hyperoxic exposure presents a potential source of damage to lung parenchyma [5, 6]. Clinically, POT presents as tracheobronchiolitis causing coughing, pleuritic chest pain, and dyspnea [7]. Clinical diagnosis is challenging due to a lack of unique objective findings: oftentimes, the only identifiable change in pulmonary function is a highly variable decrease in vital capacity (VC) [8, 9]. Other measures of pulmonary function, such as forced mid-expiratory flow and diffusion capacity (DC), have been proposed as more sensitive markers of HBOT damage, however none of these are highly specific [10].

The aim of the present study was to evaluate serial changes in pulmonary function tests among patients undergoing prolonged courses of HBOT for a variety of clinical indications. We hypothesized that extended regimens of HBOT would be associated with a degree of POT resulting in impairment of pulmonary function tests at several predetermined time intervals during a course of serial treatment.

## Methods

### Study design and participants

We conducted a retrospective analysis of prospectively collected data on a cohort of patients undergoing HBOT at the University Health Network's Hyperbaric Medicine Unit in Toronto, ON, Canada, between February 2016 and June 2021. All studied patients provided written consent to undergo HBOT (for a variety of clinical indications), and were scheduled to receive at least ten cycles of treatment at our large referral center during this timeframe. Patients underwent PFT assessment before starting HBOT and following every 20 treatment sessions thereafter.

Research ethics approval for the analysis of these data was provided by the University Health Network (Toronto, ON) Research Ethics Board (CAPCR ID: 19–5081.1). Data were collected retrospectively from the electronic records of enrolled patients, and comprised demographic information, HBOT indication and protocol, treatment complications, and PFT results immediately before the first HBOT session and following every subsequent 20 treatments. The protocol was retrospectively registered during the data collection stage and prior to analysis on Clinicaltrials.gov (trial ID: NCT05088772). We followed the STROBE guidelines for reporting observational cohort studies (S2 Table) [11].

### Hyperbaric oxygen therapy protocol

The HBOT protocol utilized at our center has been previously described [12]. HBOT was performed with 100% $O_2$ at a pressure of 2.4 or 2.0 ATA (243 or 203 kPa) for 90 minutes, with 1–2 air breaks (0.21 fraction of inspired $O_2$ at the same ATA) per session, five times weekly in one of three mono-place chambers (Sechrist 3600H and Sechrist 4100H, Sechrist Industries Inc., Anaheim, CA, USA; PAH-S1-3200, Pan-America Hyperbarics Inc., Plano, TX, USA) or

through a plastic hood in a multi-place chamber (rectangular Hyperbaric System, Fink Engineering PTY-LTD, Warana, Australia).

## Pulmonary function testing protocol

Bedside spirometry was performed by a trained respiratory therapist using a KoKo Trek USB Spirometer software and pneumotachometer (KoKo, USA). Pulmonary function tests were completed at the time of consultation (prior to the first HBOT treatment) and following every 20 treatments thereafter. In rare cases when PFTs could not be obtained on the exact date of a $20^{th}$, $40^{th}$, or $60^{th}$ treatment (e.g., due to equipment limitations), they were obtained on the nearest possible date of another treatment and rounded to an increment of 20 at the time of data analysis. The spirometry equipment was calibrated at the beginning of each day. Patients were tested in a seated position with nose clips, in accordance with American Thoracic Society testing criteria [13], and results were compared against Knudson reference values [14] to determine their percentage of predicted values based on age, sex, and height. To capture potential restrictive, obstructive, and effort-independent changes, three markers of dynamic lung function were recorded: $FEV_1$% (percentage of predicted $FEV_1$), FVC% (percentage of predicted FVC), and $FEF_{25-75}$% (percentage of predicted $FEF_{25-75}$). The data utilized in this study comprise the highest readings for each of these variables from three satisfactory forced expiratory maneuvers performed as part of each PFT assessment. The primary outcome of this study was change in spirometry performance over the course of HBOT. We additionally classified the degree of any baseline PFT abnormalities on the basis of each independent parameter's deviation from the predicted value, designating mild abnormality as 70–79%, moderate abnormality as 60–69%, and severe abnormality as less than 60%.

## Data collection and statistical analysis

Patient demographic data and past medical history characteristics were summarized using descriptive statistics, and continuous data were expressed as means ± standard deviations. Linear mixed effect regression models were used to estimate the adjusted sample mean scores of PFT outcomes $FEV_1$%, FVC%, and $FEF_{25-75}$% at each timepoint for the cohort. Timepoint was included as the fixed effect and individual subject as the random effect for each outcome for the overall cohort. PFT outcomes were also modeled for subgroups by timepoint interaction for pre-existing respiratory disease, smoking status, and treatment pressure (in ATA). Similarly, individual subjects were included as random effects. The maximum likelihood estimation was used to prepare the mixed models and analyzed under the intention-to-treat principle. Post-hoc pairwise comparisons between timepoints were conducted for each grouping of pre-existing respiratory disease, smoking status, and treatment pressure, for each PFT variable. Pairwise comparisons were adjusted using Tukey's HSD. The alpha was set to 0.05. All analyses were performed using R version 4.0.3.

## Objectives

The primary objective of this study was to evaluate changes in series of pulmonary function tests (PFTs) performed over the course of recurrent HBOT exposures. A secondary study outcome was the incidence of pulmonary complications such as lung barotrauma.

## Results

A total of 86 patients receiving HBOT during the study period were included in the analysis, all of whom received baseline spirometry and 20 or more individual treatments as illustrated

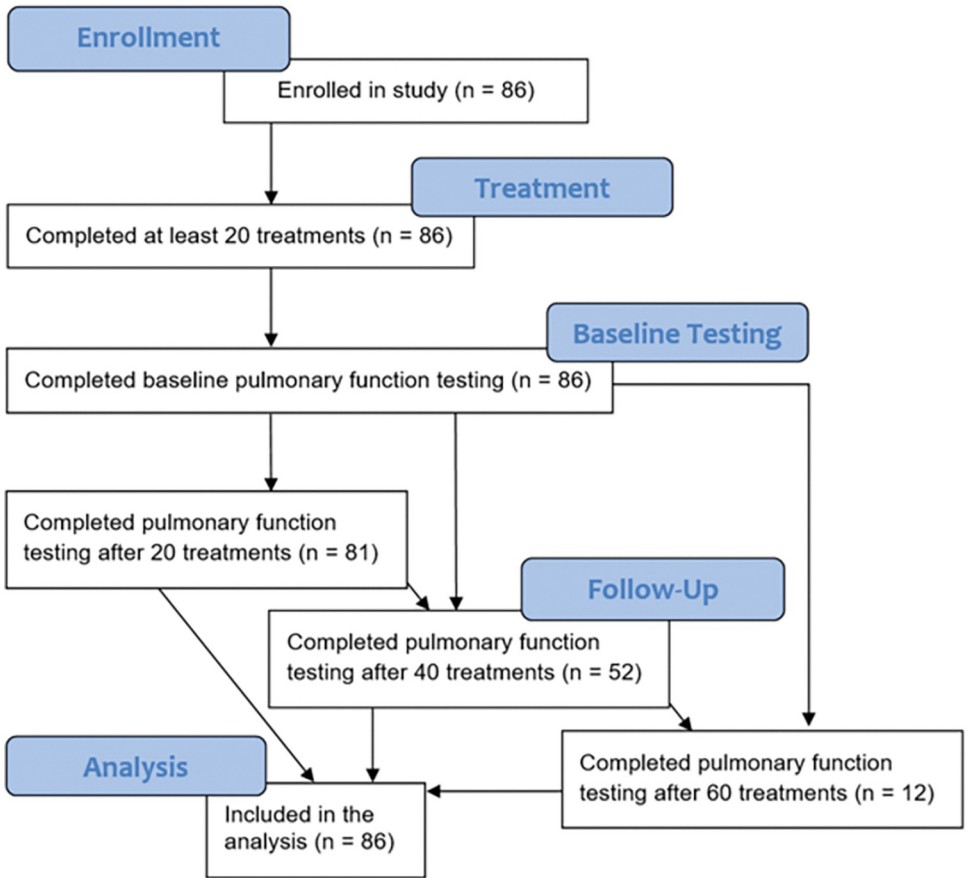

**Fig 1. Modified CONSORT diagram of enrollment.** Modified CONSORT diagram illustrating the disposition of study participants. Illustrated are the number of patients who were enrolled in the study, who underwent baseline testing, who underwent hyperbaric oxygen therapy, who underwent follow-up testing after 20, 40, and/or 60 treatments, and who were therefore including in the analysis.

in a modified CONSORT diagram [15] in Fig 1. Patients were included in the analysis if they underwent baseline spirometry prior to first treatment, as well as subsequent PFTs at one or more appropriate intervals (i.e., after 20, 40, and/or 60 treatments). A descriptive analysis of the cohort is provided in Table 1.

Patients underwent an average of 43 ± 15 (range of 20–118) HBOT sessions, detailed in Table 2. The most common indication for HBOT was soft tissue radiation injury (n = 25; 29%), and the most common complication was ear barotrauma (n = 17; 26%). No patients in the cohort experienced pulmonary barotrauma. The total number of HBOT treatments was 3666.

Breakdown of hyperbaric oxygen therapy treatment protocols for all patients included in the cohort (n = 86), including the number of cycles, the treatment pressure, and the number of air breaks, as well details regarding complications of treatment. Abbreviations: SD = standard deviation.

Due to individual variation in treatment duration, each timepoint has a unique sample size. The results of PFTs performed at baseline (n = 86) and after 20 (n = 81), 40 (n = 52), and 60 (n = 12) treatments are illustrated in Fig 2. There was no significant change in $FEV_1$%, FVC%, or $FEF_{25-75}$% across the four timepoints. A subgroup analysis comparing patients with and without pre-existing respiratory disease is presented in Fig 3. Among those with pulmonary

**Table 1. Descriptive analysis of patients included in the current study.**

| Characteristic | No. of Patients (n = 86) | Percent |
|---|---|---|
| Sex | | |
| Female | 33 | 38 |
| Male | 53 | 62 |
| Age (years) | | |
| *Mean ± SD* | 57.5 ± 15.4 | |
| Age Groups | | |
| 0–40 | 13 | 14 |
| 41–60 | 33 | 36 |
| 61–80 | 35 | 44 |
| 81+ | 5 | 6 |
| Body Mass Index* | | |
| *Mean ± SD* | 27.1 ± 10.3 | |
| Body Mass Index Groups | | |
| 15.0–18.5 | 4 | 7 |
| 18.6–25.0 | 25 | 43 |
| 25.1–35.0 | 22 | 38 |
| 35.1+ | 7 | 12 |
| Comorbidities | | |
| Hypertension | 32 | 37 |
| Coronary Artery Disease | 10 | 12 |
| Congestive Heart Failure | 5 | 6 |
| Left Ventricular Hypertrophy | 1 | 1 |
| Valvular Disease | 4 | 5 |
| Atrial Fibrillation | 5 | 6 |
| Peripheral Vascular Disease | 17 | 20 |
| Type II Diabetes | 15 | 17 |
| Chronic Obstructive Pulmonary Disease | 7 | 8 |
| Asthma | 7 | 8 |
| Interstitial Lung Disease | 2 | 2 |
| Kidney Failure | 12 | 14 |
| Previous Organ Transplant | 6 | 7 |
| Medications | | |
| Angiotensin-Converting Enzyme Inhibitor or Angiotensin Receptor Blocker | 15 | 17 |
| Beta Blocker | 13 | 15 |
| Calcium Channel Blocker | 11 | 15 |
| Diuretic | 6 | 7 |
| Digoxin | 1 | 1 |
| Opioid | 20 | 28 |
| Steroid (Systemic) | 15 | 17 |
| Anti-Platelet Agent | 19 | 23 |
| Anticoagulant | 6 | 8 |
| Beta Agonist (Inhaler) | 7 | 9 |
| Steroid (Inhaler) | 2 | 3 |
| Smoking | | |
| Current Smoker | 23 | 27 |
| Former Smoker | 16 | 19 |
| Never Smoker | 47 | 55 |

(*Continued*)

**Table 1.** (Continued)

| Characteristic | No. of Patients (n = 86) | Percent |
|---|---|---|
| Pack Years** | | |
| *Mean ± SD* | 19.5 ± 13.2 | |

Descriptive analysis of patients included in this study (n = 86), including demographic information (sex, age, body mass index), pre-existing comorbidities and medications, and smoking behavior. Abbreviations: SD = standard deviation.

*n = 58 for BMI calculations.

**n = 22 for current and former smokers for whom there was enough information to calculate pack years of smoking history.

**Table 2. Hyperbaric oxygen therapy details and complications.**

| Variable | No. of Patients (n = 86) | Percent |
|---|---|---|
| Treatment Cycles (#) | | |
| *Mean ± SD* | 42.5 ± 15.0 | |
| 20–40 | 58 | 67 |
| 41–60 | 22 | 26 |
| 61+ | 6 | 7 |
| Pressure (ATA) | | |
| *Mean ± SD* | 2.2 ± 0.2 | |
| 2.4 | 49 | 57 |
| 2.0 | 37 | 43 |
| Air Breaks (#) | | |
| *Mean ± SD* | 1.5 ± 0.5 | |
| 2 | 44 | 51 |
| 1 | 42 | 49 |
| Indication | | |
| Diabetic Foot Ulcer | 9 | 10 |
| Soft Tissue Radiation Injury | 25 | 29 |
| Osteoradionecrosis | 12 | 14 |
| Osteomyelitis | 7 | 8 |
| Idiopathic Sudden Sensorineural Hearing Loss | 3 | 3 |
| Arterial Insufficiency | 2 | 2 |
| Necrotizing Infections | 1 | 1 |
| Calciphylaxis | 2 | 2 |
| Inflammatory Bowel Disease | 2 | 2 |
| Compromised Wound | 12 | 14 |
| Treatment Complications | | |
| Ear Barotrauma | 17 | 26 |
| Lung Barotrauma | 0 | 0 |
| Seizure | 2 | 2 |
| Ocular Changes | 15 | 17 |
| Anxiety | 18 | 21 |
| Congestive Heart Failure | 1 | 1 |

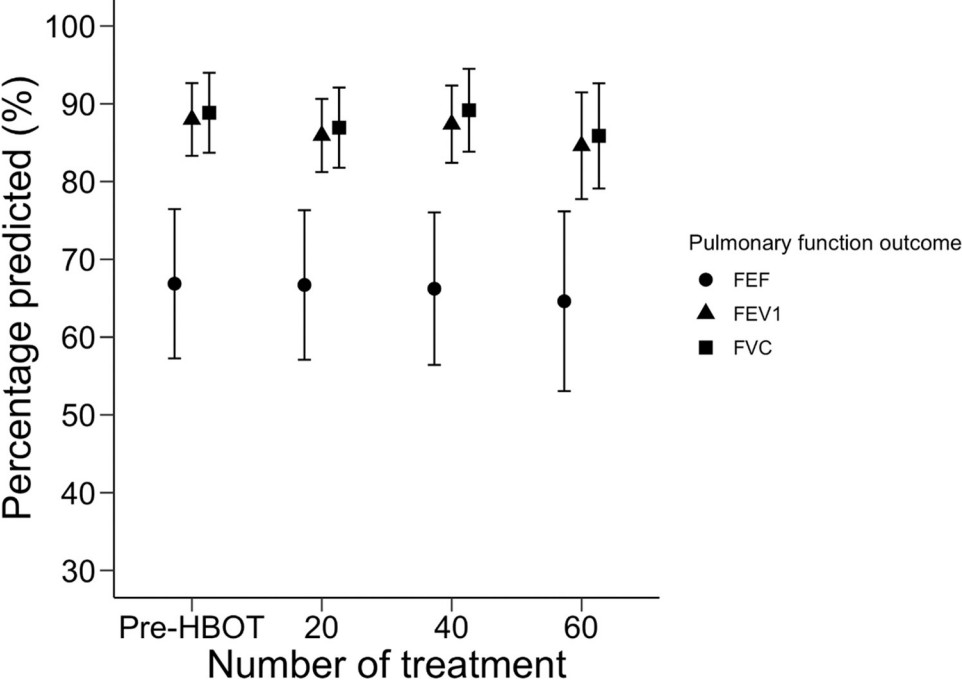

**Fig 2. Pulmonary function testing before and during hyperbaric oxygen therapy.** Measurements of pulmonary function before (n = 86) hyperbaric oxygen therapy (Pre-HBOT) and after 20 (n = 81), 40 (n = 52), and 60 (n = 12) treatment sessions. Circles, triangles, and squares represent cohort means of FEV1%, FVC%, and FEF25-75% at each timepoint, and bars delineate a confidence limit of 95%. Abbreviations: FEV1% = percentage of predicted forced expiration volume in one second; FVC% = percentage of predicted forced vital capacity; FEF25-75% = percentage of predicted mid-expiratory flow.

comorbidities, 14 patients completed PFTs at baseline, 14 after 20 treatments, and 11 after 40 treatments. No patients in this group underwent 60 treatments. Among those without pulmonary comorbidities, 72 completed PFTs at baseline, 67 after 20 treatments, 41 after 40 treatments, and 12 after 60 treatments. Neither subgroup had a significant change in $FEV_1$%, FVC %, or $FEF_{25-75}$% across these timepoints. A post-hoc pairwise comparison similarly identified no interval change in $FEV_1$%, FVC%, or $FEF_{25-75}$% values between individual study timepoints (S2 Table).

A second subgroup analysis of patients stratified by smoking status is presented in Fig 4. Pre-HBOT PFT data were available for 47 patients with no smoking history, and for 44 after 20 treatments, 31 after 40 treatments, and 10 after 60 treatments. Pre-HBOT PFT data were available for 16 patients who had formerly smoked but quit, and for 14 patients after 20 treatments, 11 patients after 40 treatments, and one patient after 60 treatments. Finally, pre-HBOT PFT data were available for 23 patients who were current smokers, and for 23 after 20 treatments, 10 after 40 treatments, and one after 60 treatments. There was no significant change in $FEV_1$%, FVC%, or $FEF_{25-75}$% across these timepoints, in any of the three subgroups. A final subgroup analysis comparing patients treated at 2.4 and 2.0 ATA (243 and 203 kPa) is presented in Fig 5. Among those treated at 2.4 ATA (243 kPa), 49 patients completed PFTs at baseline, 46 after 20 treatments, 32 after 40 treatments, and 6 after 6 treatments. Among those treated at 2.0 ATA (203 kPa), 37 completed PFTs at baseline, 35 after 20 treatments, 20 after 40 treatments, and 6 after 60 treatments. Neither subgroup had a significant change in $FEV_1$%, FVC%, or $FEF_{25-75}$% across these timepoints. Similarly, a post-hoc pairwise comparison identified no interval PFT change between timepoints in these subgroups.

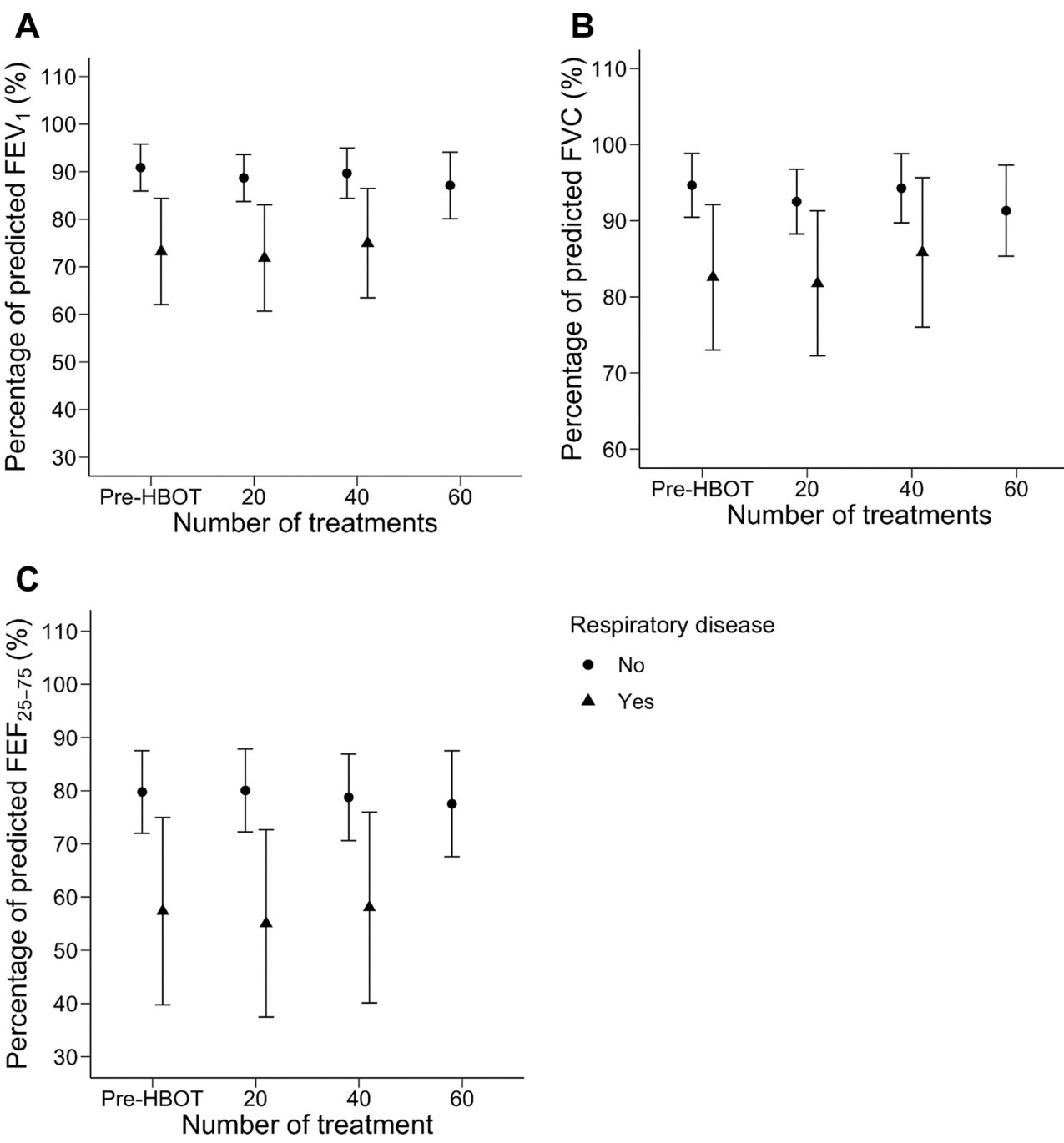

**Fig 3. Pulmonary function testing in patients stratified by pre-existing respiratory disease before and during hyperbaric oxygen therapy.**
Measurements of pulmonary function among patients stratified by pulmonary disease. Circles represent patients without known pulmonary disease before (n = 72) hyperbaric oxygen therapy (Pre-HBOT) and after 20 (n = 67), 40 (n = 41), and 60 (n = 12) treatment sessions, and triangles represent patients with pre-existing respiratory disease before (n = 14) hyperbaric oxygen therapy (Pre-HBOT) and after 20 (n = 14) and 40 (n = 11) treatment sessions. FEV1% is represented in panel A, FVC% in panel B, and FEF25-75% in panel C. Points represent subgroup means at each timepoint, and bars delineate a confidence limit of 95%. Abbreviations: FEV1% = percentage of predicted forced expiration volume in one second; FVC% = percentage of predicted forced vital capacity; FEF25-75% = percentage of predicted mid-expiratory flow.

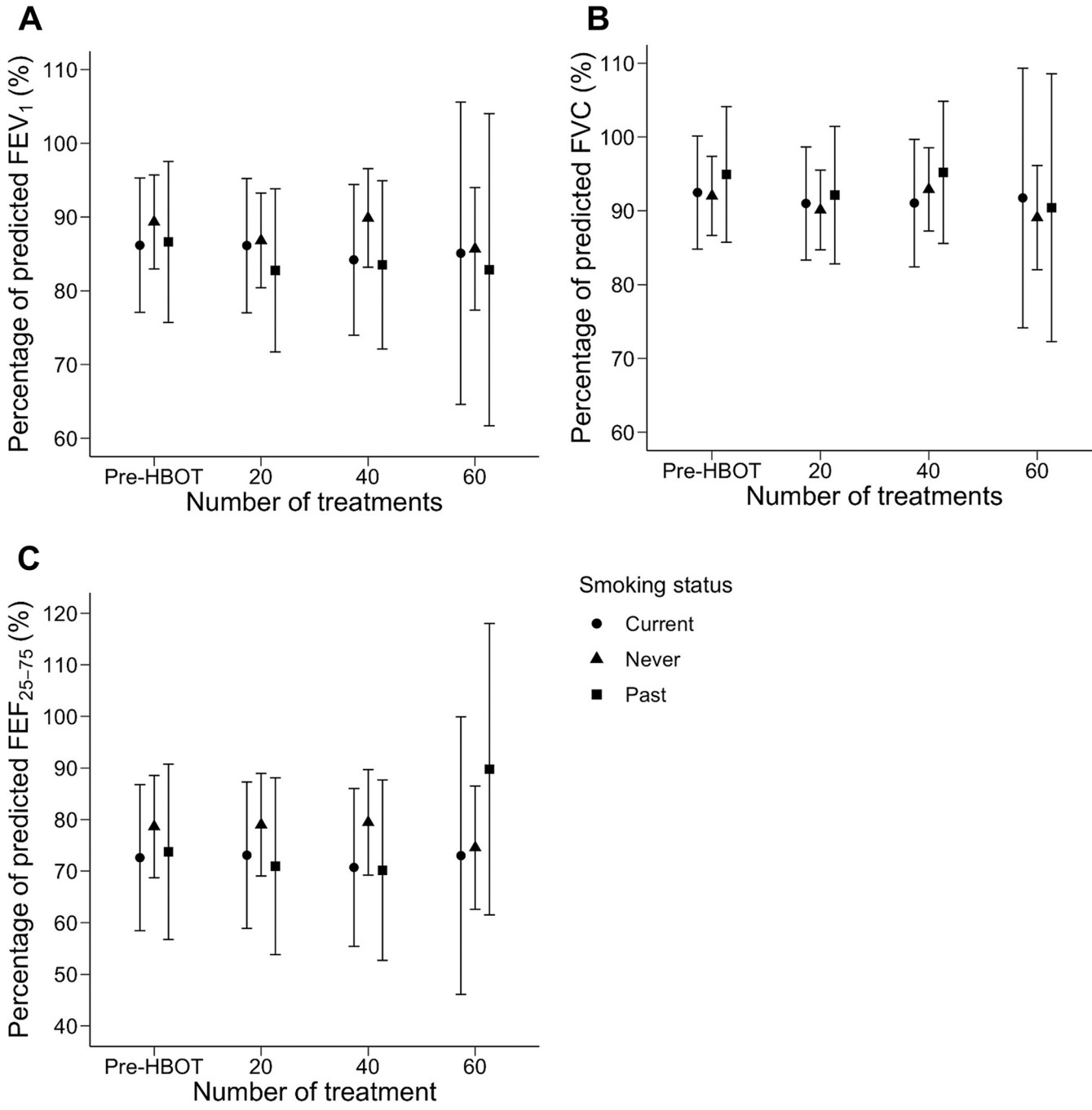

**Fig 4. Pulmonary function testing in patients stratified by smoking status before and during hyperbaric oxygen therapy.** Measurements of pulmonary function among patients stratified by smoking status. Circles represent patients who currently smoked at the time of treatment, triangles represent patients who denied any significant smoking history, and squares represent patients who formerly smoked but identified as having quit. The plots illustrate pulmonary function testing among these three subgroups, respectively, before (n = 23, 47, 16) hyperbaric oxygen therapy (Pre-HBOT) and after 20 (n = 23, 44, 14), 40 (n = 10, 31, 11), and 60 (n = 1, 10, 1) treatment sessions. FEV1% is represented in panel A, FVC% in panel B, and FEF25-75% in panel C. Points represent subgroup means at each timepoint, and bars delineate a confidence limit of 95%. Abbreviations: FEV1% = percentage of predicted forced expiration volume in one second; FVC% = percentage of predicted forced vital capacity; FEF25-75% = percentage of predicted mid-expiratory flow.

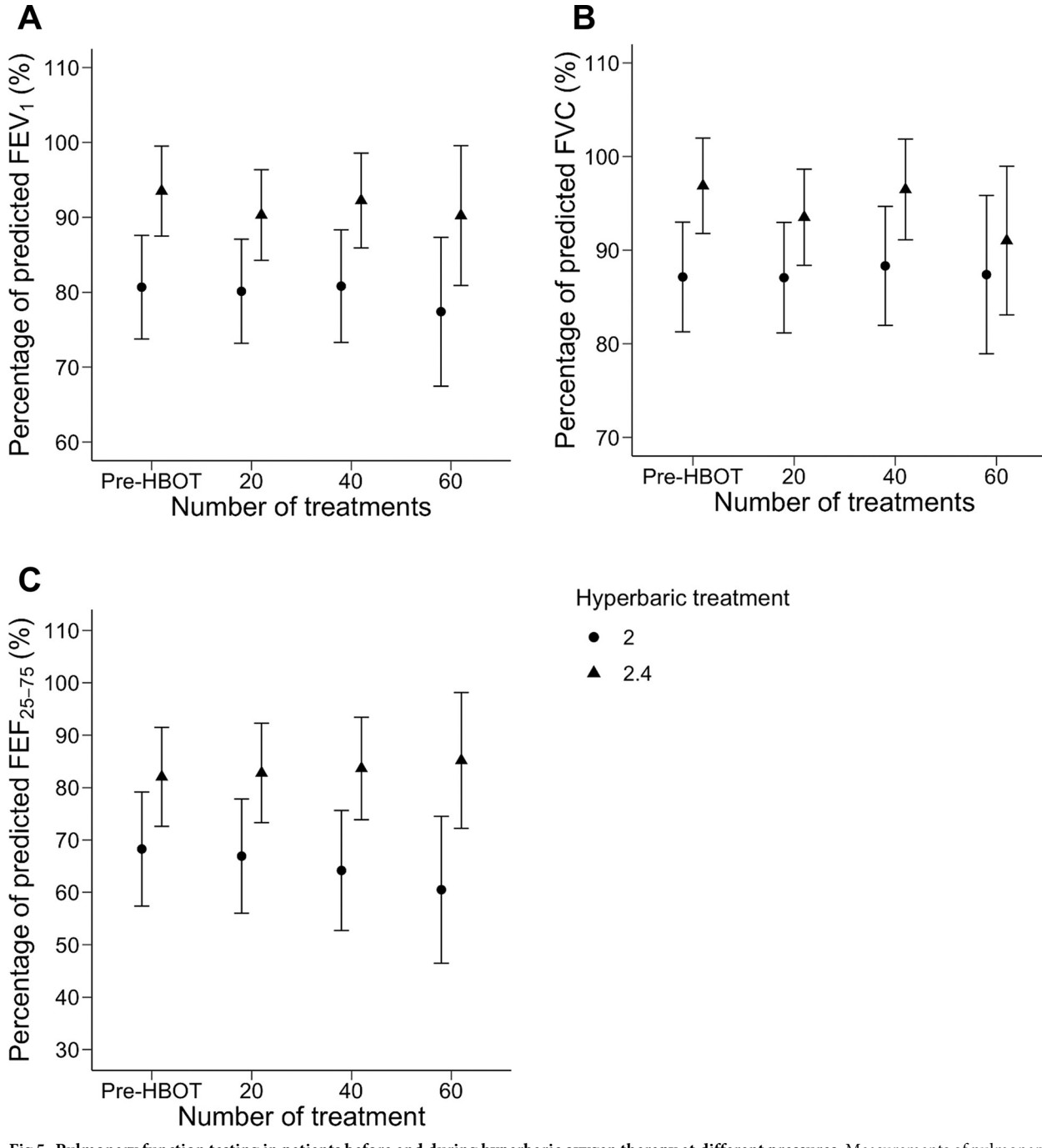

**Fig 5. Pulmonary function testing in patients before and during hyperbaric oxygen therapy at different pressures.** Measurements of pulmonary function among patients stratified by treatment pressure. Circles represent patients treated at 2.0 ATA before (n = 37) hyperbaric oxygen therapy (Pre-HBOT) and after 20 (n = 35), 40 (n = 20), and 60 (n = 6) treatment sessions, and triangles represent patients treated at 2.4 ATA before (n = 49) hyperbaric oxygen therapy (Pre-HBOT) and after 20 (n = 46), 40 (n = 32), and 60 (n = 6) treatment sessions. FEV1% is represented in panel A, FVC % in panel B, and FEF25-75% in panel C. Points represent subgroup means at each timepoint, and bars delineate a confidence limit of 95%. Abbreviations: FEV1% = percentage of predicted forced expiration volume in one second; FVC% = percentage of predicted forced vital capacity; FEF25-75% = percentage of predicted mid-expiratory flow.

## Discussion

Overall, we did not appreciate a significant change in PFTs among patients undergoing serial HBOT with a protocol of five weekly treatments of 90 minutes at 2.4 or 2.0 ATA (243 or 203 kPa) with 1–2 air breaks. This study is among the largest describing PFT changes in patients undergoing repetitive HBOT, and we report on a representative sample which is broadly generalizable to other conventional HBOT treatment facilities. Subgroup analysis identified that patients with pre-existing lung disease and those who currently or formerly smoked tended to have a greater degree of mild-to-moderate PFT abnormality at baseline; despite this, there were no significant changes in PFT trends during HBOT among these subgroups. Patients treated at 2.0 ATA (203 kPa) similarly exhibited a greater degree of mild abnormality at baseline (in all three parameters, although most markedly in $FEV_1$%, reflecting baseline differences in large airway performance). The reason for this is unclear; we speculate that providers may have elected to use more conservative treatment protocols among patients with high-risk features or whose pulmonary function already exhibited some degree of impairment prior to treatment.

Reports describing pulmonary function among human subjects undergoing HBOT are scarce, heterogenous, and divergent in their results. Pott and colleagues reported no change in forced vital capacity (FVC) or DC following 30 daily, 90-minute, uninterrupted sessions of HBOT at a pressure of 2.4 ATA (243 kPa), even among patients with significant smoking histories [16]. Thorsen and colleagues found that a treatment regimen of 21 daily, 90-minute sessions at 2.4 ATA (243 kPa), with two five-minute air breaks, considerably reduced $FEV_1$ and $FEF_{25-75}$ that did not return to baseline values [17]. In contrast, Hadanny and colleagues reported that 60 daily, 90-minute sessions at 2 ATA (203 kPa) with three, five-minute air breaks at 1 ATA (101 kPa) improved peak expiratory flow (PEF) and FVC [18]. Comert and colleagues also reported an increase in dynamic lung volumes including total lung capacity (TLC), VC, and residual volume (RV) following HBOT at 2.4 ATA (243 kPa) for 90 minutes in a cohort of 22 patients [19]. Finally, some studies have reported pulmonary function measurements among hyperbaric chamber attendants, although the frequency and duration of exposure in this population differs from that of patients undergoing HBOT. A recent observational study describes small decreases in $FEV_1$, FVC, $FEF_{25-75}$, and peak expiratory flow, with unclear clinical significance, among 68 attendants with a mean follow-up of almost ten years [20].

Our results are in agreement with those of Pott and colleagues, who similarly protocoled treatments of daily 90-minute sessions at 2.4 ATA (243 kPa), and reported no significant change in FVC after 30 treatments [16]. Our findings disagree with Thorson and colleagues who, despite following a similar protocol of 21 daily, 90-minute sessions at 2.4 ATA (243 kPa), reported a decrease in $FEV_1$ [17]. We also did not confirm the finding of Hadanny et al. or Comert et al. who reported a small increase (2.40%) in FVC% after 60 sessions of HBOT at 2.0 ATA (203 kPa) and statistically significant improvements in dynamic lung volumes with HBOT at 2.4 ATA (243 kPa), respectively (the degree of improvement, and exactly which volumes were measured, were not specified) [18, 19]. These latter studies, combined with emerging evidence based on exhaled compounds after HBOT [8], have challenged the paradigm ascribing pulmonary risks to HBOT. However, they are potentially biased by the exclusion of patients with significant pre-existing lung disease [19] or who were actively smoking [18]. The exclusion of these patients is not reflective of clinical practice, which is important as they may theoretically be among those at highest risk of pulmonary change during HBOT. The practice at our large North American referral center also differs from many of those described in these studies, frequently using treatment regimens with greater cumulative hyperoxic exposure, and the risk of POT with these protocols has not been thoroughly characterized.

The safe threshold for hyperoxic exposure in humans before risking impairment in pulmonary function appears to be approximately double the ambient air pressure at sea level (0.21 ATA or 21 kPa) [21, 22]. However, this risk is proportional to both the inspired pressure and the duration of exposure [21, 22], so that with multiple longitudinal exposures, even small elevations of ATA above that threshold may pose considerable risks. Early, exploratory studies on human subjects reported that reversible symptoms of POT and associated changes in dynamic pulmonary function develop within approximately 3–16 hours of continuous exposure to 100% $O_2$ using ATAs in a range of 1.0–3.0 (101–304 kPa) [23–26]. These changes correspond to two descriptions of two discrete phases of acute POT based on pathology of the lower respiratory tract: an acute, exudative phase characterized by reversible capillary endothelial cell damage, parenchymal edema, and the infiltration of inflammatory cells [27]; and a subacute, proliferative phase in which type II pneumocytes and fibroblasts multiply and cause irreversible derangement of the lung architecture, including marked thickening of the blood-air barrier and pulmonary fibrosis with impaired gas exchange [9, 28].

In order to quantify POT, a unit of pulmonary toxic dose (UPTD) has been introduced to predict impairment of pulmonary function [29]. As an example, hyperoxic exposures might be limited to 450 UPTD per day and 2250 UPTD per week [9] where each UPTD is the equivalent of one minute at 1 ATA (101 kPa) of 100% $O_2$. However, this model has several limitations including a need for cumulative dose calculations to account for periods of recovery between exposures, hence alternative metrics such as a POT index have been proposed [30]. Currently, there is no available metric which has been validated for modern HBOT protocols, that include daily treatment sessions at variable pressures (and with or without air breaks) over protracted intervals of time.

The UPTD of treatment protocols in our study varies with number of sessions, and between patients treated at 2.0 or 2.4 ATA (203 or 243 kPa). However, the study group with the largest exposure in our cohort (those undergoing 60 treatment sessions at 2.4 ATA or 243 kPa) would have exposure to approximately 274 UPTD per session and 16,440 UPTD in total (using the formula $UPTD = t \times [0.5/(PO_2 - 0.5)]^{-5/6}$, where $t$ is time in minutes and $PO_2$ is treatment pressure in ATA) [9]. Hadanny and colleagues calculated UPTDs for their study of 224/session and 13,489 total, as well as for the studies by Pott et al. (273/session, 8,213 total) and Thorsen et al. (273/session, 5,749 total) [18]. Using the POT index derived by Arieli [30], we calculate a safe index of 116 following an individual 90-minute treatment at 2.4 ATA (243 kPa), with essentially complete recovery over the following 22.5 hours to a negligible toxicity index of 0.001 before each subsequent treatment. Our results therefore validate the use of HBOT at both 2.0 and 2.4 ATA or 203 and 243 kPa (with a larger cumulative exposure than previously published studies), even with the inclusion of patients harboring pre-existing lung disease and/or significant smoking histories, without concerns of pulmonary dysfunction in these groups.

A secondary outcome of our study explored respiratory complications of HBOT, which are uncommon but may result from exposure of the lungs to high partial pressures of $O_2$. During the decompression phase of treatment, acute pressure changes can cause pulmonary edema [31, 32] or pulmonary barotrauma, which may lead to arterial gas embolism [33], pneumomediastinum [34], or tension pneumothorax [35], although barotraumatic lung injury is very rare in the absence of high-risk features such as pre-existing respiratory disease [36]. We did not identify any cases of pulmonary complications within our cohort, consistent with the rarity of these events reported in the current literature [37]. Our study findings therefore support the safety profile of modern HBOT.

## Limitations

While our study reports PFT measures from a large, representative cohort of patients undergoing HBOT, it is constrained by several limitations. These include a lack of DC measurement

(which could not be performed with our bedside spirometry devices), and the high degree of variability inherent in pulmonary function evaluation. A limited number of patients in our cohort missed testing at an eligible timepoint due to various resource limitations at our testing center (e.g., respiratory therapist not available to perform spirometry testing), and so may not have had PFTs performed either after 20, 40, and/or 60 treatments. Finally, our study does not include long-term follow-up to assess for possible delayed effects of HBOT on lung parenchyma and pulmonary function after the completion of treatment.

## Conclusions

The present study provides further evidence for the safety profile of HBOT, both with respect to potentially insidious consequences of treatment on pulmonary function and to acute iatrogenic injury. Our analysis of a large cohort of patients undergoing serial HBOT with periodic PFTs offers clarity to conflicting reports in the extant literature, demonstrating no significant changes in critical markers of dynamic lung function over the course of treatment. Our data also illustrate this finding in patients with prior respiratory disease or smoking histories. Future directions for this work include dose-finding studies for the safe maximum treatment pressure and duration to maximize therapeutic possibilities without impairing pulmonary function, investigations of possible delayed effects of HBOT on pulmonary function in the long term, and experiments to further characterize parenchymal changes in the hyperoxic response which may take place at the sub-clinical level.

## Supporting information

**S1 Table. Approved indications for hyperbaric oxygen therapy in Canada and the United States.** Hyperbaric oxygen therapy indications approved by Health Canada (*) or the US Food and Drug Administration (†). Unlabeled items are approved by both agencies.
(DOCX)

**S2 Table. STROBE statement for cohort studies.** Guidelines for reporting observational studies, and the page location of critical elements in the submitted manuscript.
(DOCX)

**S3 Table. Post-hoc pairwise comparisons.** Data resulting from a secondary analysis of the full cohort, which evaluated interval change in pulmonary function test performance between study timepoints. Abbreviations: FEV1% = percentage of predicted forced expiration volume in one second; FVC% = percentage of predicted forced vital capacity; FEF25-75% = percentage of predicted mid-expiratory flow; DE = difference estimate; LCL = lower confidence limit; UCL = upper confidence limit.
(DOCX)

## Author Contributions

**Conceptualization:** Connor T. A. Brenna, Rita Katznelson.

**Data curation:** Shawn Khan, Simone Schiavo.

**Formal analysis:** Connor T. A. Brenna, Darren Au.

**Methodology:** Connor T. A. Brenna, George Djaiani, Darren Au, Mustafa Wahaj, Ray Janisse.

**Project administration:** Connor T. A. Brenna, Mustafa Wahaj, Ray Janisse.

**Supervision:** Rita Katznelson.

**Visualization:** Connor T. A. Brenna.

**Writing – original draft:** Connor T. A. Brenna.

**Writing – review & editing:** Connor T. A. Brenna, Shawn Khan, George Djaiani, Darren Au, Simone Schiavo, Mustafa Wahaj, Ray Janisse, Rita Katznelson.

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
