## [Decision Letter · Decision Letter 0]

30 Jan 2023

PONE-D-22-30287Heavy breathing: a longitudinal observational study of pulmonary function following hyperbaric oxygen therapyPLOS ONE

Dear Dr. Brenna,

Thank you for submitting your manuscript to PLOS ONE. After careful consideration, we feel that it has merit but does not fully meet PLOS ONE’s publication criteria as it currently stands. Therefore, we invite you to submit a revised version of the manuscript that addresses the points raised during the review process.

 please indicate the reference for heavy breathing expression used in title as it seems a plain language rather than scientific one. The study is iteresting but what about confounders? did you considered any factors that may be associated with decreased PFT during the study period?. 

We look forward to receiving your revised manuscript.

Kind regards,

Eman Sobh, M.D.

Academic Editor

PLOS ONE

Journal Requirements:

"I have read the journal's policy and the authors of this manuscript have the following competing interests: RK is a shareholder in the Rouge Valley Hyperbaric Medical Center, Toronto, ON."

Reviewers' comments:

Reviewer's Responses to Questions

**Comments to the Author**

1. Is the manuscript technically sound, and do the data support the conclusions?

Reviewer #1: Yes

Reviewer #2: Yes

Reviewer #3: Partly

Reviewer #4: Yes

2. Has the statistical analysis been performed appropriately and rigorously? 

Reviewer #1: I Don't Know

Reviewer #2: I Don't Know

Reviewer #3: N/A

Reviewer #4: Yes

3. Have the authors made all data underlying the findings in their manuscript fully available?

Reviewer #1: Yes

Reviewer #2: Yes

Reviewer #3: Yes

Reviewer #4: No

4. Is the manuscript presented in an intelligible fashion and written in standard English?

Reviewer #1: Yes

Reviewer #2: Yes

Reviewer #3: Yes

Reviewer #4: Yes

5. Review Comments to the Author

Reviewer #1: Table1 - BMI 15-25 should be divided into under and normal weight

Table2 - ATA 48+36 = 84 implying that there are 2 missing data - please address this accordingly

Line224 "in in" - should be "in"

I wonder if we analyze interval change of PFT between time points of each subjects - will the same conclusions be drawn? (It sems to me that analyzing group data of PFT between time points maybe subject to "the stronger lives longer" kind of bias)

Reviewer #2: Thanks for conducting this important study to investigate the pulmonary function test changes in patients undergoing HBOT especially those with prior respiratory diseases or smoking histories.

I have 1 comment regarding the discussion part (265) where you stated that "Patients treated at 2.0 ATA similarly exhibited a great degree of mild abnormality at baseline" I would appreciate if you could write more details on how you categorized the PFT abnormalities. This will make it easier for the readers and other researchers too.

Reviewer #3: The subject is interesting, title is informative, aim is specific

The results support the aim and illustrative and the discussion is meticulous and answered the research question clearly.

The manuscript is well written

Reviewer #4: 1. Case Control Study were not made to determine the possible exposure factors/disease incidence

both the relative risk and odds ratio are relevant in this retrospective cohort studies.

2. The CONSORT diagram (fig-1) is not done as per the flow of data need to consider the elements of flow of subjects.

3. n=22 for smoker history. what about the others participated in this study?

and why not exclused

6. PLOS authors have the option to publish the peer review history of their article (what does this mean?). If published, this will include your full peer review and any attached files.

Reviewer #1: No

Reviewer #2: **Yes: **Abdullah A. Almojaibel

Reviewer #3: **Yes: **Aliae Mohamed Hussein

Reviewer #4: No

---

## [Author Response · Author response to Decision Letter 0]

13 Feb 2023

Editorial Team Comments:

please indicate the reference for heavy breathing expression used in title as it seems a plain language rather than scientific one. The study is interesting but what about confounders? did you considered any factors that may be associated with decreased PFT during the study period?.

Response: thank you for your comments on our manuscript. 

The original title (“Heavy breathing: a longitudinal observational study of pulmonary function following hyperbaric oxygen therapy”) is a reference to each of the core elements of the study. As noted by reviewer 3, we selected it as a concise, descriptive title for the work, introducing a more substantive subtitle which presents the methodological details of the study. Hyperbaric oxygen therapy fundamentally means breathing high-baricity (dense, pressurized, or heavy) oxygen at supra-atmospheric pressures, and the title links this concept to our primary outcome of pulmonary function/breathing performance over the course of repetitive hyperbaric exposure. With this in mind, if the editor would prefer an alternative title we are willing to modify it to: “Pulmonary function following hyperbaric oxygen therapy: a longitudinal observational study.” We have changed the title in our revised manuscript, and will leave it up to the editorial team to choose which of these two titles they prefer.

With respect to confounders, we identified key subgroups and isolated them for independent statistical analysis (including pre-existing lung disease, treatment pressure, and smoking history). Our data suggest that previous cross-sectional studies could have been confounded by these variables, for example because we see baseline differences in pulmonary function among individuals with prior lung disease that could be misinterpreted as a consequence of HBOT in a single-timepoint study. However, our longitudinal analysis did not report significant pulmonary function changes over the course of serial treatment in study group (i.e., PFT performance was stable throughout the study period), and therefore our data is not suggestive of any confounders. 

Journal Requirements:

Response: we have confirmed that our manuscript is formatted in accordance with the PLOS ONE style requirements. In particular, we have removed authors’ titles from our original submission and have ensured there are no abbreviations in our author affiliations list (Page 1). 

Response: we have clarified in our revised manuscript (Methods section, Page 4, Line 95), as well as in the online submission, that participants provided written consent for treatment (HBOT). The present study did not include minors. 

"I have read the journal's policy and the authors of this manuscript have the following competing interests: RK is a shareholder in the Rouge Valley Hyperbaric Medical Center, Toronto, ON."

We have revised our Competing Interests statement to read: 

“I have read the journal's policy and the authors of this manuscript have the following competing interests: RK is a shareholder in the Rouge Valley Hyperbaric Medical Center, Toronto, ON. This does not alter our adherence to PLOS ONE policies on sharing data and materials.”

Thank you for updating this statement in the online submission form on our behalf. 

Response: as we note in our original submission, the source data from our study cannot be shared publicly, in full, because of an ethical restriction levied by our institutional research ethics board. This is because the data contains sensitive information from patient’s medical charts (e.g., birth dates and personal health information). Taken together, this information may allow for the identification of individual study participants. We offer that an anonymized minimal data set can be prepared in aggregate and made available upon reasonable request via email to the study’s first author (connor.brenna@mail.utoronto.ca) or the Hyperbaric Medicine Unit, Toronto General Hospital, Toronto, Ontario, Canada (hyperbaricmedicineunit@uhn.ca).

Response: we have included the Ethics Statement from our online submission in the revised manuscript (please see Page 4, lines 93-98). It now reads:

“All studied patients provided written consent to undergo HBOT (for a variety of clinical indications), and were scheduled to receive at least ten cycles of treatment at our large referral center during this timeframe. Patients underwent PFT assessment before starting HBOT and following every 20 treatment sessions thereafter. Research ethics approval for the analysis of these data was provided by the University Health Network (Toronto, ON) Research Ethics Board (CAPCR ID: 19-5081.1).”

Response: we have reviewed our reference list. It is complete, and our manuscript does not cite any retracted articles. 

Reviewer #1 Comments:

Table1 - BMI 15-25 should be divided into under and normal weight

Response: we have divided the “BMI 15-25” category into “normal” (BMI 18.5-25) and “underweight” (15-18.5) as suggested by the reviewer. This change can is reflected in Table 1 (Page 8). 

Table2 - ATA 48+36 = 84 implying that there are 2 missing data - please address this accordingly

Response: thank you for identifying this typographical error. We have corrected this in the revision (it should read 49 and 37, respectively) in Table 2 (Page 10). We have also checked that the correct values were used in the subgroup analysis presented in Figure 5. 

Line224 "in in" - should be "in"

Response: thank you, this has been corrected in the revision (Page 12, Line 225).

I wonder if we analyze interval change of PFT between time points of each subjects - will the same conclusions be drawn? (It sems to me that analyzing group data of PFT between time points maybe subject to "the stronger lives longer" kind of bias)

Response: this is an interesting point about the potential for bias in interventional studies. In our study, PFT data was analyzed retrospectively after patients’ treatment was completed, and so spirometry results did not impact clinical decision making. Our study methodology therefore safeguards against this type of survivorship bias.

Nonetheless, we have performed a post-hoc pairwise comparison of the various timepoints to assess for interval change (Page 6, Lines 146-148). The results of this secondary analysis do not alter the findings or conclusion of our report, but we include this analysis of the full cohort in the revision as Supplementary Table 3. We are happy to provide the same analyses for each subgroup as additional supplemental files, if the reviewer feels it could add further value, but have not included these in the revision as each table is quite large (due to the number of variables evaluated). 

Reviewer #2 Comments:

Thanks for conducting this important study to investigate the pulmonary function test changes in patients undergoing HBOT especially those with prior respiratory diseases or smoking histories.

I have 1 comment regarding the discussion part (265) where you stated that "Patients treated at 2.0 ATA similarly exhibited a great degree of mild abnormality at baseline" I would appreciate if you could write more details on how you categorized the PFT abnormalities. This will make it easier for the readers and other researchers too.

Response: thank you for your review of our manuscript. This is an insightful comment, and we have clarified it in the revised text. We selected FEV1, FVC, and FEF25-75 as key outcome variables because they reflect large airway obstruction, lung restriction, and small/medium airway obstruction, respectively. Various classification schema exist to characterize the severity of abnormality in these variables for the purposes of clinical diagnosis, but classically rely on the interpretation of several variables in relation to each other within a single patient’s pulmonary function test. Because these variables comprise only a key subset of a full pulmonary function test, and in order to analyze them in aggregate, we opted to categorize abnormalities in FEV1, FVC, and FEF25-75 as an independent function of each parameter’s deviation from predicted values. We chose to use threshold of 70-79% to designate mild abnormality, 60-69% to designate moderate abnormality, and <60% to designate severe abnormality. However, we use these values only to comment on baseline differences in pulmonary function among subgroups; the more important outcome of our study is the stability of pulmonary function over the course of HBOT treatment, which does not appear to change significantly regardless of a patients’ starting pulmonary function. These thresholds have been clarified in the revision (Page 6, Lines 130-134), and we have added more detail to the Discussion section (Page 14, Lines 264-269) in response to this query. 

Reviewer #3 Comments:

The subject is interesting, title is informative, aim is specific

The results support the aim and illustrative and the discussion is meticulous and answered the research question clearly.

The manuscript is well written

Response: Thank you for your time and critical review of our manuscript. 

Reviewer #4 Comments:

Case Control Study were not made to determine the possible exposure factors/disease incidence

Response: a case control study would not be feasible to examine the relationship between pulmonary dysfunction and HBOT. Our data illustrates that modern HBOT protocols do not cause significant change in pulmonary function, even over long-term, repetitive treatments or among subgroups of patients who might be considered more vulnerable to pulmonary impairment. Therefore, we suggest that a case control study with modern treatment protocols could not be performed: there are no cases. 

both the relative risk and odds ratio are relevant in this retrospective cohort studies.

Response: a RR and OR cannot be calculated given that the methodology does not include a control group (e.g., without exposure to HBOT). It would be possible to calculate RR and OR metrics for variables interrogated in our subgroup analyses (e.g., treatment pressure, pre-existing pulmonary disease, and smoking history); however, because no subgroup demonstrated pulmonary dysfunction with serial treatment, the RR and OR in each of these cases would be 1 indicating an identical risk regardless of the modifying variable. In lieu or relative risk and odds ratio calculations, we present the data in Figures 2 through 5 illustrating that pulmonary function is not compromised by modern HBOT regimens. 

2. The CONSORT diagram (fig-1) is not done as per the flow of data need to consider the elements of flow of subjects.

Response: we have revised our modified CONSORT diagram (Figure 1) in the revision to better-represent the flow of subjects through the study. All patients enrolled into the study are accounted for in the updated figure, and it clearly depicts the number of patients whose enrollment continued through each study timepoint.

3. n=22 for smoker history. what about the others participated in this study?

and why not exclused

Response: 23 patients in the study were “current smokers”, 16 were “former smokers”, and 47 were “never smokers”. This comment may refer to the calculation of pack years among current and former smokers enrolled into the cohort, as precise data regarding “pack year” smoking quantity was only available for 22 enrolled patients. 

Regarding the second part of this comment, which asks why patients with smoking history were not excluded, we feel that their inclusion is one of several strengths that our present manuscript offers compared to earlier studies of the effects of HBOT on pulmonary function. We clarify in our revision (Discussion section, Page 16, Lines 296-299) that some prior studies are potentially biased by the exclusion of patients with smoking histories (or pre-existing lung disease); however, their exclusion is not reflective of contemporary practice, as a significant proportion of patients referred for HBOT have a smoking history. Thus, it is important to understand the unique risks that HBOT may bear for these patients in particular, who theoretically may be at heightened risk. For this reason, we present a subgroup analysis of patients disaggregated into the categories of current smoker, former smoker, and never smoker (Figure 4).

We thank the editor and all four reviewers for their time contributed to improving our revised manuscript.

---

## [Decision Letter · Decision Letter 1]

27 Mar 2023

PONE-D-22-30287R1Pulmonary function following hyperbaric oxygen therapy: a longitudinal observational studyPLOS ONE

Dear Dr. Brenna,

Thank you for submitting your manuscript to PLOS ONE. After careful consideration, we feel that it has merit but does not fully meet PLOS ONE’s publication criteria as it currently stands. Therefore, we invite you to submit a revised version of the manuscript that addresses the points raised during the review process.

Table 1 page 8 for the age groups and BMI the end point of each category can not be the same as the end point of the previous one (for example age groups 0-40, 40-60, ...) please revise and correct this table. 

We look forward to receiving your revised manuscript.

Kind regards,

Eman Sobh, M.D.

Academic Editor

PLOS ONE

Journal Requirements:

Reviewers' comments:

Reviewer's Responses to Questions

**Comments to the Author**

1. If the authors have adequately addressed your comments raised in a previous round of review and you feel that this manuscript is now acceptable for publication, you may indicate that here to bypass the “Comments to the Author” section, enter your conflict of interest statement in the “Confidential to Editor” section, and submit your "Accept" recommendation.

Reviewer #1: All comments have been addressed

Reviewer #2: All comments have been addressed

Reviewer #4: All comments have been addressed

2. Is the manuscript technically sound, and do the data support the conclusions?

Reviewer #1: Yes

Reviewer #2: Yes

Reviewer #4: Yes

3. Has the statistical analysis been performed appropriately and rigorously? 

Reviewer #1: I Don't Know

Reviewer #2: Yes

Reviewer #4: Yes

4. Have the authors made all data underlying the findings in their manuscript fully available?

Reviewer #1: Yes

Reviewer #2: Yes

Reviewer #4: Yes

5. Is the manuscript presented in an intelligible fashion and written in standard English?

Reviewer #1: Yes

Reviewer #2: Yes

Reviewer #4: Yes

6. Review Comments to the Author

Reviewer #1: (No Response)

Reviewer #2: (No Response)

Reviewer #4: At page 8 Table 1, need to consider the code structure of age group 0 to 40, 41 to 60, and so on. Also BMI groups.

7. PLOS authors have the option to publish the peer review history of their article (what does this mean?). If published, this will include your full peer review and any attached files.

Reviewer #1: No

Reviewer #2: No

Reviewer #4: **Yes: **ABDUL RAHMAN H ALI

---

## [Author Response · Author response to Decision Letter 1]

27 Mar 2023

Reviewer Comments

1. If the authors have adequately addressed your comments raised in a previous round of review and you feel that this manuscript is now acceptable for publication, you may indicate that here to bypass the “Comments to the Author” section, enter your conflict of interest statement in the “Confidential to Editor” section, and submit your "Accept" recommendation.

Reviewer #1: All comments have been addressed

Reviewer #2: All comments have been addressed

Reviewer #4: All comments have been addressed

2. Is the manuscript technically sound, and do the data support the conclusions?

Reviewer #1: Yes

Reviewer #2: Yes

Reviewer #4: Yes

3. Has the statistical analysis been performed appropriately and rigorously? 

Reviewer #1: I Don't Know

Reviewer #2: Yes

Reviewer #4: Yes

4. Have the authors made all data underlying the findings in their manuscript fully available?

Reviewer #1: Yes

Reviewer #2: Yes

Reviewer #4: Yes

5. Is the manuscript presented in an intelligible fashion and written in standard English?

Reviewer #1: Yes

Reviewer #2: Yes

Reviewer #4: Yes

6. Review Comments to the Author

Reviewer #1: (No Response)

Reviewer #2: (No Response)

Reviewer #4: At page 8 Table 1, need to consider the code structure of age group 0 to 40, 41 to 60, and so on. Also BMI groups.

Response: please see our earlier response, this has been corrected in Tables 1 and 2 (pages 8 and 10) per the reviewer suggestion. 

7. PLOS authors have the option to publish the peer review history of their article (what does this mean?). If published, this will include your full peer review and any attached files. Do you want your identity to be public for this peer review? For information about this choice, including consent withdrawal, please see our Privacy Policy.

Reviewer #1: No

Reviewer #2: No

Reviewer #4: Yes: ABDUL RAHMAN H ALI

Response: We thank all three reviewers for their time contributed to reviewing our revised submission to PLOS ONE.

---

## [Editor Report · Decision Letter 2]

14 Apr 2023

PONE-D-22-30287R2Pulmonary function following hyperbaric oxygen therapy: a longitudinal observational studyPLOS ONE

Dear Dr. Brenna,

Thank you for submitting your manuscript to PLOS ONE. After careful consideration, we feel that it has merit but does not fully meet PLOS ONE’s publication criteria as it currently stands. Therefore, we invite you to submit a revised version of the manuscript that addresses the points raised during the review process.

Thanks for responding to the comments. Still table one has age with the same cut off point for 0-40 and 40-60. please revise either 0-less than 40 & 40-60 or 0-40 and 41-60 and check numbers according

We look forward to receiving your revised manuscript.

Kind regards,

Eman Sobh, M.D.

Academic Editor

PLOS ONE
---

## [Author Response · Author response to Decision Letter 2]

14 Apr 2023

Editorial Team Comments

Thanks for responding to the comments. Still table one has age with the same cut off point for 0-40 and 40-60. please revise either 0-less than 40 & 40-60 or 0-40 and 41-60 and check numbers according

Response: thank you for your close attention to detail in reviewing our revised manuscript, and for identifying this error. In the further-revised manuscript, this has been corrected: the age ranges in Table 1 should have read “0-40”, “41-60”, “61-80” and “81+”. The remaining values in the table are correct without additional changes, and this error in the name of the second group has been adjusted per your suggestion. We have double-checked that our other tables and figures also represent our data correctly.

---

## [Editor Report · Decision Letter 3]

3 May 2023

Pulmonary function following hyperbaric oxygen therapy: a longitudinal observational study

PONE-D-22-30287R3

Dear Dr. Brenna,

We’re pleased to inform you that your manuscript has been judged scientifically suitable for publication and will be formally accepted for publication once it meets all outstanding technical requirements.

Kind regards,

Eman Sobh, M.D.

Academic Editor

PLOS ONE
---

## [Editor Report · Acceptance letter]

22 May 2023

PONE-D-22-30287R3 

Pulmonary function following hyperbaric oxygen therapy:a longitudinal observational study 

Dear Dr. Brenna:

I'm pleased to inform you that your manuscript has been deemed suitable for publication in PLOS ONE. Congratulations! Your manuscript is now with our production department. 

Kind regards, 

on behalf of

Dr. Eman Sobh 

Academic Editor

PLOS ONE